# Combination of Carbonate Hydroxyapatite and Stem Cells from Human Deciduous Teeth Promotes Bone Regeneration by Enhancing BMP-2, VEGF and CD31 Expression in Immunodeficient Mice

**DOI:** 10.3390/cells11121914

**Published:** 2022-06-13

**Authors:** Nurul Aisyah Rizky Putranti, Ryo Kunimatsu, Kodai Rikitake, Tomoka Hiraki, Kengo Nakajima, Takaharu Abe, Yuji Tsuka, Shuzo Sakata, Ayaka Nakatani, Hiroki Nikawa, Kotaro Tanimoto

**Affiliations:** 1Department of Orthodontics and Craniofacial Developmental Biology, Hiroshima University Graduate School of Biomedical and Health Sciences, 1-2-3 Kasumi, Minami-ku, Hiroshima 734-8553, Japan; d196538@hiroshima-u.ac.jp (N.A.R.P.); shuna-s0102@hiroshima-u.ac.jp (K.R.); tomoka1012@hiroshima-u.ac.jp (T.H.); irodori.orthod@gmail.com (K.N.); takabe@hiroshima-u.ac.jp (T.A.); tsuka1@hiroshima-u.ac.jp (Y.T.); shuzosakata@hiroshima-u.ac.jp (S.S.); anakatan@hiroshima-u.ac.jp (A.N.); tkotaro@hiroshima-u.ac.jp (K.T.); 2Department of Oral Biology and Engineering, Division of Oral Health Sciences, Institute of Biomedical and Health Sciences, Hiroshima University Graduate School of Biomedical and Health Sciences, 1-2-3 Kasumi, Minami-ku, Hiroshima 734-8553, Japan; hirocky@hiroshima-u.ac.jp

**Keywords:** bone regeneration, stem cell from human deciduous teeth, carbonate apatite, bone grafting, bone morphogenetic protein-2, vascular endothelial growth factor

## Abstract

The objective of this study was to clarify the efficiency of a combination of stem cells from human deciduous teeth and carbonate apatite in bone regeneration of calvarial defects. Immunodeficient mice (*n* = 5 for each group/4 groups) with artificial calvarial bone defects (5 mm in diameter) were developed, and stem cells from human deciduous teeth (SHEDs) and carbonate hydroxyapatite (CAP) granules were transplanted with an atelocollagen sponge as a scaffold. A 3D analysis using microcomputed tomography, and 12 weeks after transplantation, histological and immunohistochemical evaluations of markers of bone morphogenetic protein-2 (BMP-2), vascular endothelial growth factor (VEGF), and cluster of differentiation (CD) 31 were performed. In the 3D analysis, regenerated bone formation was observed in SHEDs and CAP, with the combination of SHEDs and CAP showing significantly greater bone regeneration than that in the other groups. Histological and immunohistochemical evaluations showed that combining SHEDs and CAP enhanced the expression of BMP-2, VEGF, and CD31, and promoted bone regeneration. This study demonstrates that the combination of SHEDs and CAP transplantation may be a promising tool for bone regeneration in alveolar defects.

## 1. Introduction

Regenerative medicine is a therapeutic alternative to transplantation that allows for the regeneration of dysfunctional tissues [1]. The key to cell-based regenerative medicine is stem cells, and currently, somatic mesenchymal stem cells (MSCs) are widely utilized. MSCs are unspecialized cells that can self-renew and differentiate into any other cell [2]. Advances in stem-cell biology have facilitated research into the stem-cell sources. Miura et al. reported that stem cells from SHEDs were present in primary dental pulp tissue [3]; SHEDs can be easily utilized because they can be obtained by less invasive techniques, as deciduous teeth are usually discarded [4]. Thus, our group has repeatedly focused on SHEDs in our investigations.

In previous studies, the bone-regeneration ability of SHEDs, dental pulp stem cells (DPSCs), and bone marrow MSCs (BMSCs) was at the same level in skull-deficient immunodeficient mice [5]. SHEDs also have a higher proliferative capacity than BMSCs [6]. Further, the cell population of SHEDs isolated from deciduous dental pulp exhibits a high expression of CD90, CD73, and CD105, which are positive markers of MSCs. Additionally, angiogenesis and bone-regeneration ability were higher in CD146-positive SHEDs than in heterogeneous SHED cells and CD146-negative SHED cells [7]. However, when bone defects such as cleft palates are present, the 3D morphology of the defect is complicated, and external mechanical loading occurs during oral function. Therefore, the use of combined carriers, which are artificial materials, is essential in cell transplantation for bone regeneration in the maxillofacial region.

Using a material with a composition similar to that of natural bone minerals is important for reducing inflammatory responses and achieving optimal resorption behavior [8]. Artificial bone materials such as hydroxyapatite and β-tricalcium phosphate composites (β-TCP) are applied in oral surgery and prosthodontic fields [9,10]. These inorganic materials have high affinity for osteoblasts but are difficult to clinically evaluate for bone regeneration because they are mostly non-resorbable or have long resorption times and are radiopaque. Further, non-absorbable carriers may inhibit tooth eruption and the artificial movement of teeth. Therefore, a carrier in which regenerated bone acquires early physiological bone metabolism is desirable. Unsintered carbonate apatite (CAP), wherein a part of the hydroxyapatite crystal structure is replaced by carbonic acid, is the main inorganic component of bone and teeth and has high bio-affinity and shape retention. Additionally, containing carbonic acid, CAP is highly soluble in acid, and is susceptible to resorption by osteoclasts [11]. The resorption properties of CAP could be attributed to the tendency of carbonates to reduce crystallinity in the apatite structure; this promotes bone remodeling or turnover. CAP undergoes only osteoclastic resorption; therefore, its absorption rate closely matches that of natural bone [12].

CAP promotes osteoblast differentiation and CAP artificial bones have high bone conductivity [12,13]. A comparison between CAP bone prostheses and autologous bone revealed that CAP was similarly absorbed and replaced by bone in osteoclast precursor cell-culture experiments and in a rat tooth extraction socket model [14]. In addition, CAP bone prostheses showed promising results in the biopsy–histological assessment upon implant entry and three years after implantation in two-way cases of maxillary sinus floor elevation [15,16].

We conducted a pilot study focusing on the usefulness of CAP [17]. Preliminary studies clarified that in a beagle canine jaw fissure model, bone regeneration occurs in the jaw row division after BMSCs and CAP carrier transplantation; the transfer of the tooth to the bone-regeneration position is possible [17,18,19]. However, bone regenerative capacity and regenerated bone assessment have not been compared among SHEDs, CAP, and combined SHEDs and CAP transplantations, and the mechanism remains unclear. Therefore, this study aimed to compare bone-regeneration ability and evaluate bone regeneration after SHEDs, CAP, and combined SHEDs + CAP transplantations in immunodeficient mice with skull bone defects. We aimed to build a scientific rationale, based on the obtained results, for the use of this therapeutic strategy to promote optimal bone regeneration in cleft lip/palate.

## 2. Materials and Methods

### 2.1. Ethics Statement

Human deciduous teeth pulp harvest was approved by the Preliminary Review Board of the Epidemiological Research Committee of Hiroshima University (approval no. E-20-2). All experimental protocols were approved by the Ethics Committee for Animal Experiments of Hiroshima University School of Dentistry (approval no. A 20-81).

### 2.2. Cell Isolation

Upper right primary canine teeth were extracted from 11-year-old male patients undergoing orthodontic treatment at Hiroshima University Hospital; the SHEDs were isolated and cultured. Informed consent was obtained from the parents of all donors. After extraction, teeth were immediately soaked in phosphate-buffered saline (PBS) containing 100 mM amphotericin. The periodontal ligaments were dissected after the teeth were disinfected with isodine and hibitane. The teeth were split at the cementoenamel junction using an osteoclamp. Pulp tissue was then collected and dissected in 10 mL α-MEM with 4 mg/mL collagenase and 3 mg/mL dispase. Subsequently, it was transferred to 10 mL tubes containing collagenase and dispase solution and incubated at 37 °C and 5% CO_2_ for a maximum of 30 min.

The tubes were then centrifuged for 5 min at 1500 rpm. The supernatant was aspirated, and the tissue was suspended in α-MEM with 20% (*w*/*v*) fetal bovine serum (FBS), 0.24 µL/mL kanamycin, 0.5 µL/mL penicillin, and 1 µL/mL amphotericin. The suspension was then cultured in a 35 mm cell-culture dish and incubated at 37 °C and 5% CO_2_. When at least 200 colonies were formed, the cells were removed from the culture dish, PBS containing 0.25% (*w*/*v*) trypsin and 1 mM ethylenediamine tetra-acetic acid was added, and the cells were passaged. After the first passage, the cultures were incubated at 37 °C and 5% CO_2_ in 10% (*w*/*v*) FBS/α-MEM with the aforementioned antibiotics [5,6,7]. Cells from the sixth passage were used in this study.

### 2.3. Calvarial Bone Defect Immunodeficient Mouse Model

To prevent immunogenic and graft rejections, 6-week-old male immunodeficient mice (BALB/c-nu; Japan Charles River International Laboratories Inc., Yokohama, Japan) were used. Mice were fed non-fluorescent, alfalfa-free, solid food. An anesthetic consisting of midazolam, medetomidine, and butorphanol was applied before surgery. After each mouse was administered general anesthesia, a 5.0 mm-diameter calvaria defect was made using a trephine bur in the center of the calvaria, as previously described [5,7]. CAP (Cytrans Granules; GC Corporation, Tokyo, Japan) was ground using a nano-grinder (NP-100; Thinky Corporation, Tokyo, Japan) to produce a mean particle size of ~110 nm. The mean particle size was validated by scanning electron microscopy (SEM) (S-3400N, Hitachi High-Technologogies, Tokyo, Japan) and particle-size analysis (scattering intensity and laser diffraction).

The graft material containing SHEDs and 110 nm-sized CAP was implanted in the defect area using an atelocollagen sponge (Mighty; ⌀ 5.0 × 1.5 mm; Koken, Tokyo, Japan). Four groups (5 mice/groups) were established based on the implanted material: (a) control, serum-free α-MEM (25 μL/atelocollagen sponge); (b) SHEDs (10^5^ cells/atelocollagen sponge); (c) CAP (50 μL CAP/atelocollagen sponge); (d) SHEDs + CAP (10^5^ cells + 50 μL CAP/atelocollagen sponge). Eight weeks after transplantation, the mice were sacrificed for histological analysis.

### 2.4. Microcomputed Tomography (μCT) Analysis

The calvarial area was scanned by μCT (Skyscan1176; Bruker, Kontich, Belgium) at a resolution of 35 μm immediately after transplantation (t0) and at 4 (t1) and 8 weeks (t2). The scans were performed parallel to the coronal aspect of the calvaria. ZedView (Lexi, Tokyo, Japan) was used for 3D reconstructions of the microdiographic images, and RapidForm 2006 (INUS Technology, Seoul, Korea) and FreeForm (SensAble Technologies; Wilmington, MA, USA) were used, respectively, to cut and measure the images. Bone volume was measured by calculating the difference between the filled spaces at t1 and t2 in the 3D constructed defect of each group.

### 2.5. Histological Evaluation via Hematoxylin and Eosin (H&E) and Masson’s Trichrome (MT) Staining

After the mice were sacrificed, the regenerated tissue was fixed in 4% (*w*/*v*) paraformaldehyde PBS, soaked for decalcification in 14% EDTA for 1 month, fixed in paraffin, and then sectioned into 5 μm-thick sections. H&E and MT staining were performed as previously described [5,7]. In MT staining, tissue sections were viewed under a fluorescence microscope (BZ-X800, Keyence, Osaka, Japan) coupled with BZ-II imaging analysis software (Keyence). A range was selected in the tissue section in which the 5 mm-diameter defect was created. The areas stained red indicated mature bone, and the areas stained blue indicated collagen fibers. These along with the osteoid were measured using BZ-II imaging analysis software (Keyence).

### 2.6. Immunohistochemical (IHC) Analysis

IHC analysis was performed to detect bone morphogenetic protein-2 (BMP-2) expression. For the analysis of angiogenesis on the slice from the center of the transplant site, sections were stained for vascular endothelial growth factor (VEGF-A) and CD31 markers. After deparaffinization and dehydration of the tissue sections, Dako Protein Block was used to inhibit non-specific responses. Anti-VEGF-A antibody, anti-CD31 antibody, and anti-BMP-2 antibody were used as primary antibodies and allowed to react overnight at 4 °C. The primary antibodies were diluted in sterile PBS. After the cells were washed with PBS, they were treated with anti-rabbit IgG as a secondary antibody for 1 h at room temperature. The color was then developed with 3,3′-diaminobenzidine using a Histofine^®^ SAB-PO kit. Subsequently, sections were counterstained with hematoxylin. For VEGF-A and BMP-2 IHC staining, tissue sections were observed using a fluorescence microscope (BZ-X800, Keyence) and a range of 5 mm defects were selected in the tissue sections. The percentage of VEGF-A- and BMP-2-stained areas in the transplanted area was calculated using the BZ-II imaging analysis application (Keyence). For CD31 IHC staining, CD31-positive blood-vessel counts in four random ranges were calculated at 200× magnification to evaluate angiogenesis. Tissue sections were observed using fluorescence microscopy (BZ-X800, Keyence) and analyzed using the BZ-II imaging analysis application (Keyence).

### 2.7. Statistical Analysis

The Kruskal–Wallis test was performed using Bell Curve for Excel (SSREI). The differences among groups were analyzed using the Bonferroni method. Data are presented as mean ± standard deviation (SD). Significance was considered at *p* < 0.05 and *p* < 0.01.

## 3. Results

### 3.1. SEM Analysis and 3D Evaluation of Regenerated Bone after In Vivo Transplantation with SHEDs + CAP, SHEDs, or CAP

The SEM analysis of CAP granules showed a mean particle size of 110 nm at 100× magnification (Figure 1a).

At t0, in all groups, no new bone formation was observed in the calvarial area. In the control group at t1 and t2, wound closure at the bone defect area was seen, but only a few newly regenerated bones were observed. At the center of the calvarial defects in the other groups, significant wound closure with newly generated bone was observed relative to that in the control group (Figure 1c). The SHEDs + CAP group had significantly greater bone volume at four and eight weeks after transplantation compared to that of the other groups. Additionally, significantly greater bone volume was observed in the CAP and SHEDs groups than in the control group (Figure 1d).

### 3.2. Comparison of Histological Evaluation Results

#### 3.2.1. H&E Staining

In the H&E staining, newly formed bone was clearly observed in the SHEDs, CAP, and SHEDs + CAP groups. In contrast, only a few newly formed bone areas were observed in the control group. Biodegradation of the scaffold is shown as a blank area in Figure 2a.

#### 3.2.2. MT Staining

In the MT staining, areas stained red indicated mature bone, and areas-stained blue indicated collagen fibers and osteoids. In the control group, only a few collagen fibers and osteoids were observed. In contrast to the control group, the SHEDs + CAP group prominently exhibited collagen fibers, osteoids, and newly formed bone. In the SHEDs group, a small amount of newly formed bone was observed (Figure 2b). Among transplantation sites, the percentage area of mature bone was significantly higher in the SHEDs + CAP transplantation group than in other groups. The SHEDs and CAP transplantation groups showed significantly enhanced percentage areas of mature bone compared to the control group (Figure 2c).

#### 3.2.3. IHC Staining

IHC staining for BMP-2, VEGF and CD31 was performed to observe bone formation and angiogenesis, respectively.

##### Comparison of BMP-2 Expression in SHEDs, CAP, and SHEDs + CAP Groups

The SHEDs + CAP group had a prominent BMP-2-stained area at the scaffold center (Figure 3a). The percentage of the area stained for BMP-2 among transplanted sites was significantly higher in the SHEDs + CAP transplantation group compared to the other groups (Figure 3b). Compared to the control group, the SHEDs transplantation group showed a significantly higher percentage area of BMP-2 (Figure 3b). No significant differences in BMP-2 expression were observed between the CAP transplantation and control groups (Figure 3b).

##### Comparison of Angiogenesis in SHEDs, CAP, and SHEDs + CAP In Vivo Transplantation

After IHC staining for VEGF-A, the SHEDs + CAP transplantation group showed a significantly enhanced percentage of the VEGF-A area in the transplantation site relative to the other groups (Figure 4a,b). Additionally, the CAP transplantation and SHEDs transplantation groups showed significantly higher proportions of the VEGF-A area compared to the control group (Figure 4b). After IHC staining for CD31 expression, a significant increase in new blood vessels was observed in the SHEDs + CAP group, but only a few new blood vessels were observed in the control group (Figure 5a). There were significantly more blood vessels in the SHEDs + CAP group (10.85 ± 1.76 blood vessels); however, no significant difference was found in the number of blood vessels between the SHEDs and SHEDs + CAP groups. SHEDs, CAP, and SHEDs + CAP groups had significantly more blood vessels than the control group (Figure 5b).

## 4. Discussion

Transforming apatite carriers into smaller-sized granules may facilitate transplantation and lead to earlier bone and carrier resorption. Therefore, in this study, CAP granules were crushed to an average size of 110 nm, and CAP transplantation was performed in calvarial defects in immunocompromised mice. μCT analysis at one and two months after transplantation showed significantly more bone regeneration in the CAP transplantation group than in the control group. Additionally, the equivalent bone-regeneration image was accepted, although significance was not observed in the SHEDs and CAP transplantation groups. Moreover, marked mature bone formation was confirmed in the CAP transplantation group by MT staining two months after transplantation. The residue of the CAP carrier was not confirmed. In recent studies, CAP promoted osteoblast differentiation and CAP artificial bone had high bone conductivity [13]. Comparing CAP bone prostheses with autologous bone revealed that CAP was similarly absorbed and replaced by bone in osteoclast precursor cell-culture experiments and a rat tooth extraction socket model [14]. Additionally, good results were reported from the biopsy–histological assessment in CAP bone prostheses upon implant entry and three years after implantation in two-way cases of maxillary sinus floor elevation [15,16]; our results are similar to previous results.

Osteogenesis in MSCs can be efficient with sustained stimulation by osteo-inductive bio-factors, such as BMP-2. BMP-2 promotes bone formation by directing MSC differentiation into osteoblasts or osteocytes [20,21]. Our previous in vitro study demonstrated that SHEDs have a higher BMP-2 expression compared to BMSCs or DPSCs [6]. Based on these findings, we investigated SHEDs transplantation combined with CAP and the consequent changes in BMP-2 expression. The SHEDs + CAP transplantation group clearly showed significantly higher bone regeneration than the control, CAP, and SHEDs transplantation groups. Additionally, BMP-2 IHC staining showed that the percentage of areas stained for BMP-2 among the transplantation sites was significantly higher in the SHEDs + CAP transplantation group than in the control, SHEDs, and CAP transplantation groups. Moreover, compared to the control group, the SHEDs transplantation group had significantly higher BMP-2 expression. A recent study showed that transplantation in Wistar rats using SHEDs-CAP scaffolds can enhance BMP-2 and BMP-7 expression, while the attenuation of MMP-8 expression strongly indicated that SHEDs-CAP scaffolds may be promising treatment for bone regeneration [22]. CAP scaffolds have osteoinductive and osteoconductive properties that can induce a microenvironment suitable for SHEDs to differentiate into an osteogenic lineage [22]; this is consistent with our findings. However, we found no significant differences in BMP-2 expression between the CAP transplantation and control groups. Here, we discuss the possibility of developing BMP-2 from transplanted SHEDs and its effects on transplanted tissues. The effects of BMP-4, BMP-6, and other growth factors such as basic fibroblast growth factor (bFGF) should be investigated and elucidated.

VEGF is a signaling protein that regulates angiogenesis, which consists of four stages, refers to the formation of new blood vessels, and occurs throughout life. VEGF plays a dual role in this process: First, it acts on endothelial cells to promote migration and proliferation, and second, it stimulates osteogenesis through regulation of osteogenic growth factors [23]. In the present study, the expression levels of VEGF and CD31 were significantly higher in the SHEDs and SHEDs + CAP groups than in the other two groups. This is consistent with the results of a previous study wherein dental-derived MSCs secreted a range of proangiogenic factors, including VEGF, FGF2, and platelet-derived growth factor (PDGF) when endothelial cells received pro-angiogenic signals, which bind to their corresponding receptors on endothelial cells [24].

BMP-2 and VEGF are important factors involved in endothelial cell and osteoblast coordination and are mediated by multiple growth factors and cytokines for bone formation [25]. The VEGF and BMP-2 combination has been studied for its interaction with angiogenesis and osteogenesis in MSCs [26]. The formation of supportive vascular networks can be stimulated by VEGF, which affects BMP-2 and enhances bone formation [27]. Additionally, VEGF may function as a mobilization cytokine for endothelial progenitor cells, which promote bone regeneration. Endothelial cell and osteoblast migration from neighboring tissues can be stimulated by VEGF and BMP-2 released from scaffolds [28]. Therefore, BMP-2, VEGF, and bFGF expression may be regulated by reciprocal signaling pathways and, via these interactions, may result in the synergistic promotion of MSC osseous differentiation [29]. Therefore, in vitro studies of SHEDs and CAP should be conducted in the future to elucidate their signaling pathways in terms of bone regenerative action mechanisms.

Our study suggests that, compared to SHEDs or CAP transplantation alone, SHEDs + CAP transplantation can enhance BMP-2 and VEGF expression and induce superior angiogenic and osteogenic potential. However, with the aim of elucidating SHEDs and the clinical applications of SHEDs and CAP, various factors must be studied, including cell number, cell population homogeneity, growth factors, CAP granule size, and detailed bone-regeneration mechanisms.

## 5. Conclusions

This study led to the following conclusions: (1) The transplantation of SHEDs with CAP granules can induce more new bone formation than the transplantation of either SHEDs or CAP alone, and (2) the transplantation of SHEDs with CAP granules can enhance BMP-2 expression and promote angiogenesis by enhancing VEGF and CD31 expression. These results indicate that the SHEDs + CAP combination may be a promising tool for bone regeneration in alveolar defects.

## Figures and Tables

**Figure 1 cells-11-01914-f001:**
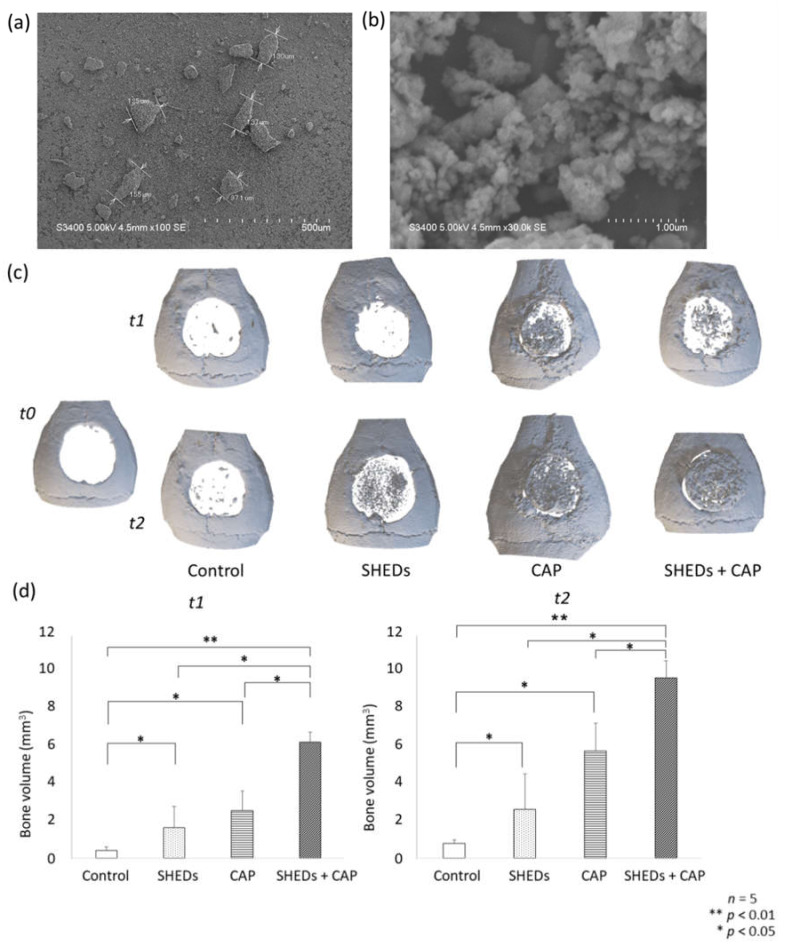
SEM analysis of nanostructured CAP granules showing the mean size of CAP particles. (**a**) 100× magnification (scale bar = 500 μm). (**b**) 30,000× magnification (scale bar = 1.00 μm) μCT analysis from 3D images of regenerated bone. No newly formed bone was observed at t0 in any group (**c**). At t1 (4 weeks), a small amount of newly formed bone was detected in the control and SHEDs groups, and newly formed bone was more prominent in the CAP and SHEDs + CAP groups than in the control group. Newly formed bone volume was significantly higher in the SHEDs, CAP, and SHEDs + CAP groups at t1 and t0 (**d**).

**Figure 2 cells-11-01914-f002:**
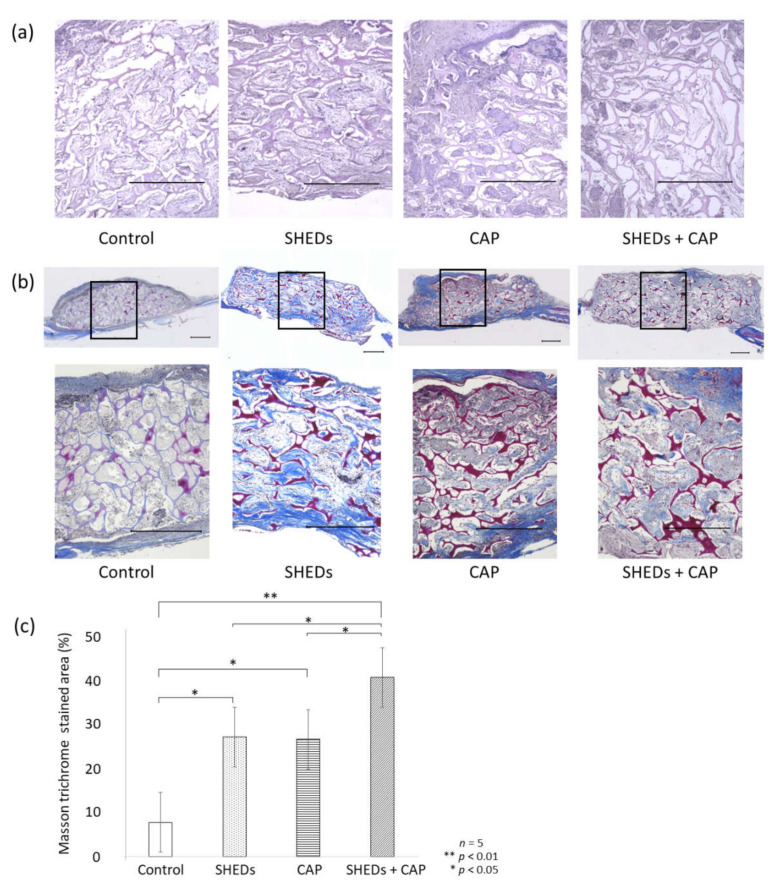
Histological evaluation of regenerated bone area. (**a**) Post H&E staining, the control group showed only a few calcified-like tissues, whereas the SHEDs + CAP group exhibited many such tissues. (**b**) MT staining showed a clear staining site for mature bone in the SHEDs, CAP, and SHEDs + CAP groups but not in the control group. (**c**) Percentage of stained area was significantly higher in the SHEDs + CAP group than in the other groups. *n* = 5 for each group, ** *p* < 0.01, * *p* < 0.05. Scale bars = 500 μm.

**Figure 3 cells-11-01914-f003:**
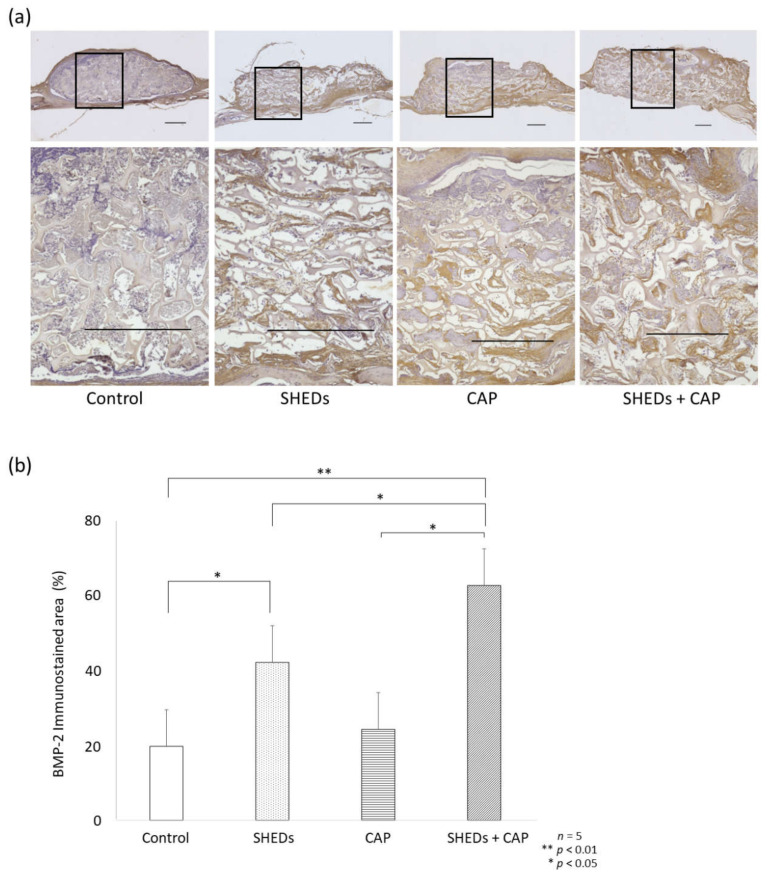
IHC analysis of the expression of BMP-2. (**a**) IHC analysis of BMP-2 expression showed that only a few sites of expression were observed in the control group but were more clearly observed in the other groups. (**b**) Statistical analysis of the stained area showed a significantly higher BMP-2 expression in the SHEDs and SHEDs + CAP groups but not in the control group or CAP group. *n* = 5 for each group, ** *p* < 0.01, * *p* < 0.05. Scale bars = 500 μm.

**Figure 4 cells-11-01914-f004:**
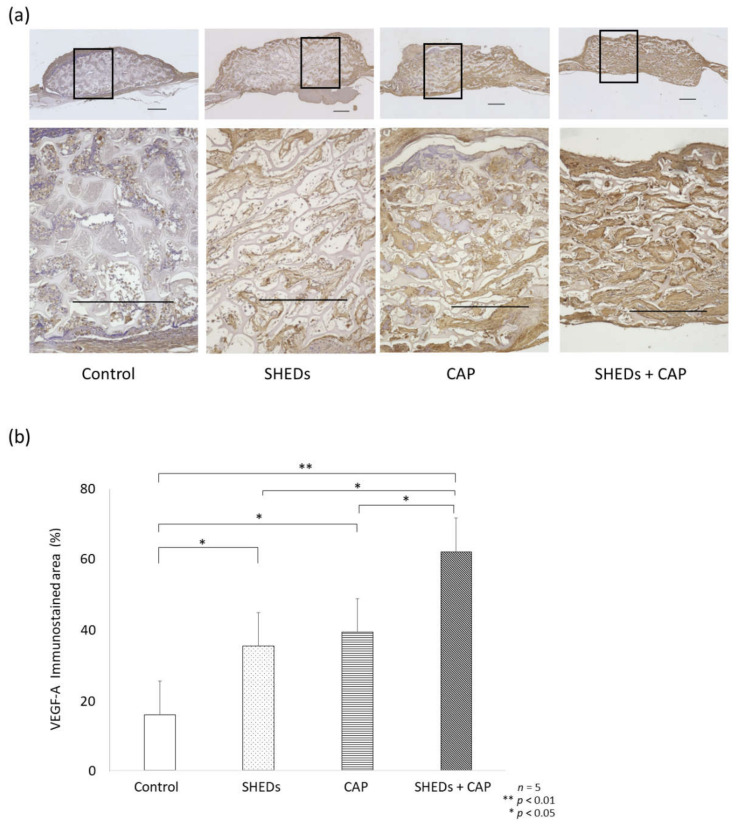
IHC analysis of VEGF. (**a**) IHC analysis of VEGF expression showed clearer and more vividly stained areas in the SHEDs, CAP, and SHEDs + CAP groups than in the control group. (**b**) The proportion of the VEGF-stained area was significantly higher in the SHEDs, CAP, and SHEDs + CAP groups than in the control group. *n* = 5 for each group, ** *p* < 0.01, * *p* < 0.05. VEGF scale bar = 500 μm.

**Figure 5 cells-11-01914-f005:**
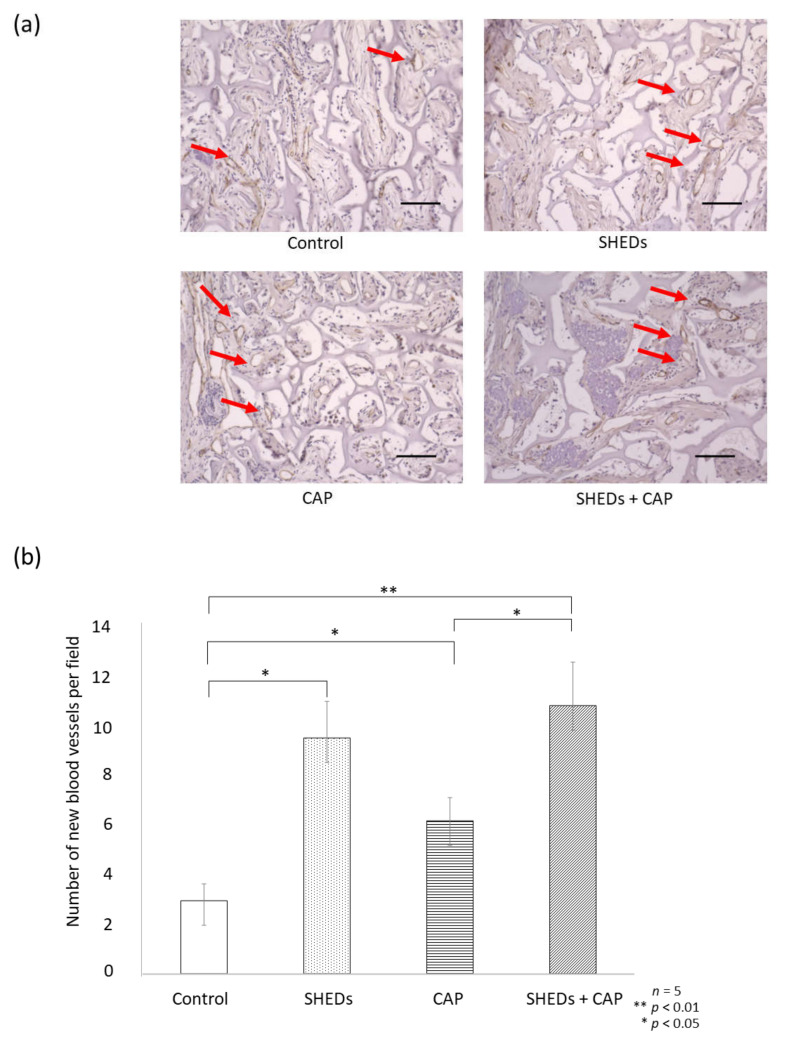
IHC analysis of CD31. (**a**) IHC analysis of CD31 expression revealed new blood vessels (indicated using red arrows) in the SHEDs, CAP, and SHEDs + CAP groups compared to the control group. (**b**) SHEDs + CAP had significantly more blood vessels than the CAP group and the control group. There was no significance difference in the number of blood vessels between the SHEDs and SHEDs + CAP groups. *n* = 5 for each group, ** *p* < 0.01, * *p* < 0.05. CD31 scale bar = 100 μm.

## Data Availability

The data presented in this study are available on request from the corresponding author.

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
