# Peer review of "Combination of Carbonate Hydroxyapatite and Stem Cells from Human Deciduous Teeth Promotes Bone Regeneration by Enhancing BMP-2, VEGF and CD31 Expression in Immunodeficient Mice"

_cells, 2022, doi:10.3390/cells11121914_

Round 1
Reviewer 1 Report
Dear Respected Authors,
The manuscript entitled “Combination of carbonate apatite and stem cells from human deciduous teeth promotes bone regeneration by enhancing BMP2, VEGF, and CD31 expression in immunodeficient mice “ tries to compare bone regeneration ability and evaluate bone regeneration after SHED, CAP, and combined SHED+CAP transplantations in immunodeficient mice with skull bone defects. The manuscript is overall well-written with careful and thorough searching endeavors. However, some comments are given below to improve your manuscript:
1- It is highly suggested that one form of a word be used throughout the whole text, for instance, what does CAP stand for? Carbonated hydroxyapatite or Carbonate apatite. What is written in the title and abstract section doesn’t match the one in the introduction section.
2- Abstract section, Page 1: Please kindly use the abbreviation form of the following terms after their first use in the abstract section in order to decrease the redundancy: Use SHEDs and CAP instead of stem cells from human deciduous teeth and carbonate apatite, respectively.
3- Introduction section, Page 2: The manuscript jumps almost immediately into the aim of the study; therefore, there is a need for at least a paragraph for the review of the current literature regarding the relevant studies. I kindly recommend the authors amend this.
4- Materials and Methods section, Page 3, Calvarial bone defect immunodeficient mice model, Last sentence of the first paragraph: It was written in the text that although the mean particle size was validated by scanning electron microscopy and particle size analysis, there is no data for showing the results. I am wondering why there aren't any results regarding the validation of the mean particle size by SEM and particle size analysis.
5- Materials and Methods section, Page 3, Microcomputed tomography (μCT) analysis: The last sentence indicates that there is a t3 time point in the study which I think was a misspelling. Please kindly amend this as well; otherwise, explain more about it.
6- Results section, Page 7, Immunohistochemical staining: I highly suggest the authors omit the following sentence from the 3.2.3.1. section and add to the 3.2.3. section, since the following sentence is a general statement and is not specific to the BMP-2 section.
· IHC staining for BMP-2 and for VEGF and CD31 was performed to observe bone formation and angiogenesis, respectively.
Author Response
June 6th, 2022
Cells
Dear reviewer 1,
We wish to re-submit the manuscript titled, “Combination of carbonate hydroxyapatite and stem cells from human deciduous teeth promotes bone regeneration by enhancing BMP2, VEGF, and CD31 expression in immunodeficient mice”. The manuscript ID is [Manuscript ID cells-1753494].
We thank you for the valuable comments and suggestions. Accordingly, the manuscript has been rechecked and the necessary changes have been made throughout the manuscript. The point-by-point responses to the comments have been prepared and attached herewith. The revisions made in response to the comments are marked using yellow highlights, in the revised manuscript.
We look forward to working with you to move this manuscript closer to publication in the Cells. Thank you for your consideration. We look forward to hearing from you.
Sincerely,
Ryo Kunimatsu

Reviewer 2 Report
This is an interesting paper about bone regeneration. It is well written and high in novelty. I have few remarks:
Apstract- the number of mice should be given.
Introduction- the references 8 and 9 are too old, give more updated ones (i.e. https://doi.org/10.1016/j.actbio.2020.06.022, https://doi.org/10.1515/bmt-2019-0218).
Materials- how many teeth/patients were used for the study- were there any differences in their osteogenic capacity of the SHED, how were the cells characteriszed to being SHED?
The authors should explain in the text why was the concentration of 100.000 cell was used for the regeneration experiments.
Lines 133, 134- the authors should explain the method of measurment or cyte appropriate reference.
Figures: the quality of the pictures should be improved. Give explanation of the red arrows in fig. 5a in figure legend.
Author Response
June 6th, 2022
Cells
Dear reviewer 2,
We wish to re-submit the manuscript titled, “Combination of carbonate hydroxyapatite and stem cells from human deciduous teeth promotes bone regeneration by enhancing BMP2, VEGF, and CD31 expression in immunodeficient mice”. The manuscript ID is [Manuscript ID cells-1753494].
We thank you for the valuable comments and suggestions. Accordingly, the manuscript has been rechecked and the necessary changes have been made throughout the manuscript. The point-by-point responses to the comments have been prepared and attached herewith. The revisions made in response to the comments are marked using yellow highlights, in the revised manuscript.
We look forward to working with you to move this manuscript closer to publication in the Cells. Thank you for your consideration. We look forward to hearing from you.
Sincerely,
Ryo Kunimatsu

Round 2
Reviewer 1 Report
-
Reviewer 2 Report
No further changes are needed.